# WIPO initiative of transforming traditional resources and sharing rights: An evolutionary game analysis and a Chinese context

**Dong Zhang**[1]*, **Rui Huang**[2], **Chaoran Lin**[3]

1 College of Marine Culture and Law, Jimei University, Xiamen, P.R. China, 2 College of Marxism, Harbin Engineering University, Harbin, P.R. China, 3 School of Economics and Management, Harbin Engineering University, Harbin, P.R. China

* zhangdongipr@jmu.edu.cn

## Abstract

WIPO-GRTKF specifies, for the first time, how traditional resources embodied by traditional knowledge, genetic resources, and folklore can be defined, and what the relationship between original rights, and rights arising from the transformation and utilisation of traditional resources can be understood. Committed to promoting innovation, shared benefits and balanced interests, WIPO tries to achieve a balance between preventing users from violating holders' original rights through the acquisition of patent, trademark and copyright, and incentivizing all stakeholders to transform traditional resources to improve the greater good. The document triggers a new round of disputes among interest groups over how to share rights arising from the transformation and utilisation of traditional resources. Using an evolutionary game model to simulate how holders and users transform traditional resources, and share rights, we find that when the two sides choose to cooperate to apply for transformation and give consent to use, their benefits are maximised and strategies stabilised. We suggest that in the transforming process, holders' rights and users' interests be given equal emphasis, and an autonomous and open mode combining statutory licensing, and justified utilisation of original rights be employed. We advocate for a hybrid legislative arrangement that integrates the incentive of IPRs as private rights, and the safeguard of public rights. In the dual subject system, both users and holders enjoy multiple rights in the process of protecting and transforming traditional resources. The Chinese approach to transforming traditional resources and sharing their rights will contribute to sustainable development of traditional resource industry across the globe.

## Introduction

How to achieve justified sharing of rights arising from the transformation of traditional resources has attracted attention from scholars of economics and intellectual property rights (IPRs). In June 2023, WIPO Intergovernmental Committee released the Draft on Genetic Resources, Traditional Knowledge and Folklore (WIPO-GRTKF-IC-47), representing its

**Data Availability Statement:** All data files are available from the GitHub database website via the link below: https://github.com/huangruiipr/Traditional-Resource-Evolution-Game.git.

**Funding:** DZ; DHA230***; Ministry of Education of the People's Republic of China; http://onsgep.moe.

edu.cn/edoas2/website7/index.jsp; The funders had no role in study design, data collection and analysis, decision to publish, or preparation of the manuscript.

**Competing interests:** The authors have declared that no competing interests exist.

attempt to resolve issues around traditional resources embodied by traditional knowledge (TK), genetic resources (GR) and folklore (F), along with the relationship between original rights and the rights arising from transformation and utilisation. Member states are given autonomy to establish rights-sharing systems conducive to transformed traditional resources accessible to global audiences. In other words, the document provides non-binding solutions that countries can adapt to their own conditions. It is only natural that countries may have diverse interpretations of the WIPO guidelines, in both theoretical and practical dimensions. The interpretation and practice of China, a country rich in traditional resources, will have a profound impact on others' approaches to dealing with the issue. China's Outline for Protecting Intellectual Property Rights (2021-2035) stipulates that the country will "build a robust system facilitating the access and benefit-sharing of genetic resources, traditional knowledge, and folklore, and strengthen the collection, classification, transformation and utilization of intangible cultural heritage" [1]. However, the Outline does not stipulate how to share rights arising from the transformation. This paper analyses disputes over WIPO-GRTKF, presents an optimal solution to transforming traditional resources and sharing rights from an evolutionary game theoretic perspective.

## Background

Article 8 of Preamble/Introduction in WIPO/GRTKF/IC/47/4 reads, "Acknowledging that the protection of traditional knowledge should contribute toward the promotion of creativity and innovation, and to the transfer and dissemination of knowledge to the mutual advantage of holders and users in a manner conducive to social and economic welfare and to a balance of rights and obligations" [2]. This provision clarifies the objectives of WIPO-GRTKF to transform, utilise, inherit, and develop traditional resources. These objectives deserve a thorough study, because they present historic solutions, spark scholarly debate, and cover a multitude of dimensions.

The concept of "traditional resource rights" was proposed by the non-governmental organization Global Coalition for Biocultural Diversity. In 1992, the United Nations adopted the Convention on Biological Diversity (CBD), proposing the notion of "access and benefit sharing (ABS)". The Agreement on Trade-Related Aspects of Intellectual Property Rights (TRIPS) initiated by the United States in 1994 accelerated the imbalance of benefits arising from the transformation and utilisation of TK, and the advancement of patent applications in other countries [3]. Published in January 2022, Article 11.53 in the Regional Comprehensive Economic Partnership Agreement (RCEP) gives a detailed explanation of the core components of traditional resources. In other words, the three elements of Genetic Resources, Traditional Knowledge, and Folklore, are reaffirmed as the inherent attribute of traditional resources and as the object of protection and utilisation.

However, none of the above-mentioned international treaties has clearly defined the sharing of traditional resource rights. On the one hand, although TK, F, and GR have cultural and property values, most of the existing international treaties only stipulate the transformation and utilisation of GR, ignoring the rights arising from the transformation and utilisation of TK and F. Both CBD and its supplementary agreement, Nagoya Protocol on Access to Genetic Resources and the Fair and Equitable Sharing of Benefits Arising from their Utilization (NP), define the object of protection as GR and TK. Though the RCEP does cover all the three components, their protection is not as comprehensive as expected. For the protection of GR, the document requires that the parties disclose the source or origin of genetic resources and pursue quality patent examination. But the protection of TK and F is only strengthened through quality patent examination. On the other hand, neither the TRIPs Agreement nor the RCEP

clarifies the meaning of justified sharing of original rights and the rights arising from the transformation of traditional resources, nor mentions the benefit-sharing principle, an innovative strategy started by the CBD.

## Academic contentions

First, scholars remain divided on the relationship between IPRs and the sharing of rights arising from the transformation of traditional resources. Being intangible property created for transmitting cultural and spiritual values, the transformation and utilisation of traditional resources are related to private rights, which are protected by the legal system of IPRs. However, this system does not explicitly explain how to protect traditional resources as a form of intellectual property (IP). There is a marked difference in determining subject and protection duration between IP and traditional resource rights. For example, international treaties such as the CBD, NP and RCEP, regional agreements, and the Intangible Cultural Heritage Law of the People's Republic of China have not clearly defined the subject of traditional resource rights. According to the Paragraph 1, Article 2 of the Copyright Law of the People's Republic of China, the subject of copyright is "Chinese citizens, legal persons or unincorporated organizations." Similarly, according to Paragraph 1, Article 42 of the Patent Law of the People's Republic of China, "The duration of patent right for inventions shall be twenty years, the duration of patent right for utility models shall be ten years, the duration of patent right for designs shall be fifteen years, counted from the date of filing." However, since traditional resources are created, accrued and inherited by indigenous communities, it is difficult to determine the protection duration, and start and end time. Therefore, both theories on and practical solutions to the protection of rights arising from the transformation of traditional resources deserve further examination.

Second, academics have not reached a general consensus on the game relationship and approaches to sharing rights arising from the transformation and utilisation of traditional resources. Paterson argues that rights of indigenous people have distinctive features and cannot be equated with ownership [4]. Hilty maintains that developed regions aim to profit from private rights out of collectively owned traditional resources, which are being exploited by foreign competitors beyond the control of developing countries [5]. It is fair to say that the relationship between private rights and collective rights, and the game conditions for interest groups to share rights are decisive factors that shape rights-sharing approaches in the process of transforming and utilising traditional resources.

Third, legal guarantee to share traditional resource rights has also been a source of contention. Some scholars believe that the transformation and utilisation of traditional resources should be achieved through a benefit-sharing system. Anchored in the concept of exchange justice [6] the existing benefit-sharing system can produce desired results if international and national initiatives complement each other [7]. However, other scholars are not so optimistic. They argue that the success of benefit-sharing system is far from satisfactory. Access to traditional resources such as GR is limited, and the system is inadequate [8–10], necessitating more sophisticated forms, such as taxes or levies [11].

Fourth, though Chinese researchers have made initial achievements in protecting IPRs of traditional resources, research on transformation is scarce. In 2021, the Chinese government issued the Opinions on Strengthening the Protection of Intangible Cultural Heritage, formulating the principle of protecting traditional resources in terms of "prioritising protection and rescue, and valuing justified utilisation, inheritance and development." China places greater emphasis on protection than on transformation and utilisation. This stance can be explained from three perspectives. The first draws on the Single Private Right Theory, which maintains

that the existing IPRs framework is sufficient to protect traditional resources [12–14]. The second draws on the Independent Public Right Theory, which states that to clarify the collective attribute of traditional resource rights, researchers need three legal norms, namely, "informed consent", "origin indication" and "interests sharing" [15, 16]. The third perspective draws on the Hybrid Theory, which combines IPRs and a special system. Wu believes that a system dedicated to traditional resource rights distinguishes itself from the present IPRs law, without undermining the very foundation of international IPRs protection system [17]. He argues, "It is encouraging to see that a brand-new rights system for traditional resource rights is attracting attention from academics around the world. This system is aimed at protecting traditional property and original property, both of which serve as a fountain of creativity for modern IP."

To sum up, while a consensus has been reached on the implementation of international treaties and the application of benefit-sharing system, disagreements exist in priority setting. Chinese researchers focus more on how to protect traditional resources and which legal system to apply, but less on the sharing of rights arising from the transformation of traditional resources, and still less on the conceptualization of rights-sharing approaches from an interdisciplinary perspective. We believe that China's current IPRs system is not suitable for sharing traditional resource rights. New developments in how stakeholders share rights under the WIPO-GRTKF framework merit scholarly attention and innovative approaches that take into account local conditions are called for. In this study, we propose a new system to balance protection, transformation, and utilisation. This balanced approach, we argue, in combination with new breakthroughs represented by WIPO-GRTKF, will provide a pragmatic solution to sharing rights arising from the transformation of traditional resources.

## Document analysis

As mentioned above, WIPO/GRTKF/IC/47 gives the definition of traditional resources embodied by TK, GR and F, as well as the relationship between original rights and the rights arising from the transformation and utilisation of traditional resources. Compared with the TRIPs Agreement, CBD and RCEP, this draft elaborates on concepts related to traditional resource rights in a more constructive way and builds a comprehensive framework for protecting original traditional resources. Founded in 2001, WIPO Intergovernmental Committee on Intellectual Property and Genetic Resources, Traditional Knowledge and Folklore (henceforth IGC) was founded in 2001. Since its founding, IGC had not given a clear definition of how to protect traditional resource rights until June 2023 when WIPO/GRTKF/IC/47) was released. This document represents four major breakthroughs. First, the number of elements consisting of traditional resources that can be transformed and utilised is expanded. In the revised document, the previous requirement of "use and utilisation" that is "outside the traditional scope" has been removed, and would-be transformed resources include those within the traditional framework. Second, the definition of traditional resource holders is further clarified. The previous provision of "indigenous (individuals), local communities and/or (other beneficiaries)" has been changed to "indigenous individuals and local communities" to distinguish these entities from users of traditional resources [18]. Third, the draft further acknowledges the importance of respecting indigenous individuals and local communities and strengthens the protection of their property rights. Users need to obtain prior informed consent of TK holders before using it for transformation and utilisation. If users utilise TK for invention and creation, they should disclose the origin when applying for patents from competent authorities in WIPO Member States. Fourth, WIPO-GRTKF prevents users from violating holders' original rights through the acquisition of patent, trademark, and copyright, as well as incentivizes all stakeholders to creatively transform traditional resources to promote the greater good. Equal

emphasis is given to the protection of original traditional resource rights and the transformation rights.

New technologies give stronger impetus to the inheritance and development of worldwide traditional resources, necessitating the reframing of subject which engages in transforming and utilising the resources. To some extent, to transform traditional resources involves commercialising them. Identifying the components of subject is a prerequisite for the commercialisation. Considering that WIPO-GRTKF is dedicated to promoting sustainable innovation and public wellbeing, we identify holders and users as the subject of traditional resource rights. In WIPO/GRTKF/IC/47/4, Articles 9 and 14 of the Preamble, and Articles 1, 2 and 5 of the main body stipulates the protection of beneficiaries' rights. Article 8 of the preamble reads, "Acknowledging that the protection of traditional knowledge should contribute toward the promotion of creativity and innovation, and to the transfer and dissemination of knowledge to the mutual advantage of holders and users in a manner conducive to social and economic welfare and to a balance of rights and obligations." This clause illuminates a game relationship between holders and users of TK. Rather than being antagonistic, the two sides should work together for mutual advantage, balance interests, promote the inheritance and development of traditional resources in a sustainable manner, and contribute to the creation, innovation, transfer and dissemination of knowledge. The drafters have paid more attention to the balance of interests between holders and users, which has a direct bearing on continuous transformation and dissemination of traditional resources. They have realized that there is nothing wrong with identifying users as the subject of traditional resource rights, who can engage in the process of transformation and utilisation as arrangers, performers, adaptors, and re-creators/creators. For example, Article 9.6 (b) in WIPO/GRTKF/IC/46/4 states, "the [protected] traditional knowledge was obtained from one or more holders of the [protected] traditional knowledge with their free, prior and informed consent or approval and involvement." Although it stipulates users' obligation to obtain prior informed consent from one or more holders of protected TK, the article fails to deal with the feasibility of how the holders, be they communities, nations, or countries, obtain prior informed consent.

## Materials and methods

To apply the principle of respect and balanced sharing advocated by WIPO-GRTKF to the above-mentioned game relationship, we need to find a solution for holders and users to share rights arising from the transformation of traditional resources. Built on bounded rationality, evolutionary game theory takes groups, rather than individuals, as research object, and uses continuous game as evidence. Therefore, this theory can be applied to explain the game of rights-sharing related to the transformation of traditional resources. We clarify that our study did not involve any activities that would require ethical review and approval. Our research did not involve human participants or animals, and no personal or sensitive data was used. Therefore, it was not necessary to seek approval from an Institutional Review Board (IRB) or ethics committee. However, we understand the importance of ethics in research and assure you that our study was conducted in accordance with the general ethical guidelines of our field.

### Theoretical logic: Phase and symbol of evolutionary game simulation

Evolutionary Game Model might be applied to explain the evolution of traditional resource rights. This paper attempts to present an evolutionary game model to describe the game process between users and holders who share and maximize their respective traditional resource rights. To conduct the game matrix analysis, we base our assumption on the following

preconditions: subject, strategy, implication of choosing the consent strategy, and symbols for matrix logic, i.e. costs and benefits.

**Phase assumption.** *(1) Game subjects.* The evolutionary game model involves two game subjects, that is, users and holders in the process of sharing traditional resource rights. To be more specific, indigenous communities, natural persons and groups are holders. Users include natural persons, enterprises and countries that produce, import, promise to sell, sell, store, or utilise products of traditional resources, or research and develop these resources.

*(2) Strategy.* Considering that the sharing of traditional resource rights is in line with benefit-sharing systems and voluntary agreements between holders and users, holders may choose to consent or not when users apply for access to and utilization of traditional resources.

*(3)* Costs and benefits of consent and application. Holders' consent affects whether they themselves will invest in skill transfer, and database building, and whether users will illegally utilise traditional resources, and reap branding benefits generated by authentic traditional resources. Users' application impacts whether they are willing to bear application costs, and whether holders can derive additional benefits arising from the utilisation of traditional resources.

**Symbol description.** *(1) Costs.* When sharing traditional resource rights, users shall bear three types of costs: input costs ($C_1$) for transforming and re-creating, application fee ($C_2$), and costs of illegal use if holders disagree ($C_3$). In the same process, holders shall bear the costs of transferring skills ($C_4$) and building databases ($C_5$).

*(2) Benefits.* There are four types of benefits, namely, total income from the utilisation, transformation and re-creation of traditional resource rights ($R_1$); additional benefits obtained by users who apply to promote traditional resources, and build brands out of authentic traditional resources ($R_2$); additional benefits obtained by holders ($R_3$), since more individuals, enterprises and countries know about traditional resources and seek opportunities for cooperation; and third-party funds obtained by holders and users who agree to share with each other in accordance with a rights-sharing agreement ($R_4$).

*(3) Probability.* If the proportion of users who choose to apply is *x*, then the proportion of users who choose not will be $1 - x$. If the proportion of holders who choose to consent is *y*, then the proportion of holders who choose not will be $1 - y$. $0 \le x \le 1, 0 \le y \le 1$.

*(4) Other symbols.* *p* is the ratio agreed by users to share interests in a benefit-sharing agreement. $1 - p$ signifies the ratio agreed by holders in the same agreement. $0 \le p \le 1$.

## Empirical analysis: Evolutionary game presenting holders and users' selection of approaches to transforming traditional resources and sharing rights

We analyze the benefit matrix in which both players choose different strategies, construct a replicator dynamics equation, and find five equilibrium points. Then the Jacobian matrix is employed to calculate whether each point is ESS, and which conditions are needed to reach this state. Our analysis shows that when certain conditions are met, there is a long-term stable set involving traditional resource holders and users in the evolutionary game, and that changes in costs and benefits can increase the probability of the two sides choosing the rights-sharing mode.

**Modelling evolutionary games.** Based on the above assumptions, when users and holders play a game in sharing traditional resource rights, the following four game modes can be produced. The game benefit matrix of holders and users is shown below (Table 1).

Based on the benefit matrix in Table 1, we draw conclusions from the research of Bach et al (2006) [19] to explore time-evolving characteristics of users' approach selection, which leads to

**Table 1. Benefit matrix.**

| HoldersUsers | Consent | Not consent |
|---|---|---|
| Apply | $(pR_1 + R_2 + R_4 - C_1 - C_2, (1-p)R_1 + R_3 + R_4 - C_4 - C_5)$ | $(R_1 - C_1 - C_2 - C_3, R_3)$ |
| Not apply | $(R_1 + R_2 - C_1, -C_4 - C_5)$ | $(R_1 - C_1 - C_3, 0)$ |

replicator dynamics equation indicating user application selection, as shown in Eq (1).

$$dx/dt = F(x) = x(1-x)(pR_1 y - R_1 y + R_4 y - C_2) \tag{1}$$

The benefit matrix in Table 1 can also lead to the replicator dynamics equation to signify how holders choose to consent in Eq (2).

$$dy/dt = F(y) = y(1-y)(R_1 x - pR_1 x + R_4 x - C_4 - C_5) \tag{2}$$

**Stability analysis of evolutionary game models.**   To analyse the best evolutionary path for game subjects who participate in the transformation of traditional resources and share their rights in the game system, researchers can take the following three steps. First, assigning $F(x) = 0$, $F(y) = 0$ to calculate the equilibrium point in the game system where traditional resources are transformed and their rights are shared. Second, drawing on the research done by Luthie et al (2005) [20] to construct replicator dynamics Jacobian matrix in the evolutionary game model indicating game subjects who transform traditional resources and share rights. Third, calculating determinant value and trace value at each equilibrium point, and judge whether each point is ESS in line with the evolutionarily stable strategy proposed by Friedman (1998). If the determinant (Det) of the matrix corresponding to the equilibrium point is greater than zero, and the trace (Tra) less than zero, it is ESS. If the trace equals zero, then it is a saddle point [21, 22].

Friedman's method inspires us that the stability of equilibrium point can be achieved by analysing the local stability of the Jacobian matrix in Eq (3).

$$J = \begin{bmatrix} (1-2x)(pR_1 y - R_1 y + R_4 y - C_2) & x(1-x)(pR_1 - R_1 + R_4) \\ y(1-y)(R_1 - pR_1 + R_4) & (1-2y)(R_1 x - pR_1 x + R_4 x - C_4 - C_5) \end{bmatrix} \tag{3}$$

Determinant of the Jacobian matrix in Eq (4).

$$detJ = (1-2x)(pR_1 y - R_1 y + R_4 y - C_2)(1-2y)(R_1 x - pR_1 x + R_4 x - C_4 - C_5) - \\ y(1-y)x(1-x)(pR_1 - R_1 + R_4)(R_1 - pR_1 + R_4) \tag{4}$$

Trace of the Jacobian matrix in Eq (5).

$$trJ = (1-2x)(pR_1 y - R_1 y + R_4 y - C_2) + (1-2y)(R_1 x - pR_1 x + R_4 x - C_4 - C_5) \tag{5}$$

Five equilibrium points calculated through the Jacobian matrix are (1, 0), (0, 0), (1, 1), (0, 1) and $(x^*, y^*)$, among which $x^* = \frac{C_4 + C_5}{(1-P)R_1 + R_4}$ and $y^* = \frac{C_2}{R_4 - (1-P)R_1}$. Local stability analysis shows that (1, 0) and (0, 1) are unstable points, $(x^*, y^*)$ is saddle point, (0, 0) and (1, 1) are evolutionarily stable ESS in Table 2.

**Discovery of game dynamic phase.**   It can be seen in Fig 1 that when $R_4 - C_2 > (1-p)R_1 > C_4 + C_5 - R_4$, there are two stable evolutionary strategies: (0, 0) and (1, 1). When the strategy converges to (0, 0), it indicates that in most cases, users and holders do not cooperate. When the strategy converges to (1, 1), users and holders choose a benefit-sharing mode and form a

**Table 2. Stability analysis results of the Jacobian matrix.**

| Condition | (0, 0) | (0, 1) | (1, 0) | (1, 1) | $(x^*, y^*)$ |
|---|---|---|---|---|---|
| $R_4 - C_2 > (1-p)R_1 > C_4 + C_5 - R_4$ | ESS | Unstable point | Unstable point | ESS | Saddle point |
| $(1-p)R_1 < R_4 - C_2\ (1-p)R_1 < C_4 + C_5 - R_4$ | ESS | Unstable point | Unstable point | Unstable point | Saddlepoint |
| $(1-p)R_1 > R_4 - C_2\ (1-p)R_1 > C_4 + C_5 - R_4$ | ESS | Unstable point | Unstable point | Unstable point | Saddlepoint |
| $C_4 + C_5 - R_4 > (1-p)R_1 > R_4 - C_2$ | ESS | Unstablepoint | Unstable point | Unstable point | Saddle point |

stable partnership. The polygonal OADC area signifies that holders and users choose not to consent or apply, converging to (0, 0); the polygonal ABCD shows the opposite, converging to (1, 1); and the size of the polygon symbolizes the possibility that the two sides choose the corresponding strategy. The larger the value, the greater the probability that users and holders will choose rights-sharing mode. Considering that A, B, O, and C are all fixed points, the area of polygon ABCD depends on the position of D $\left(\frac{C_4+C_5}{(1-P)R_1+R_4}, \frac{C_2}{R_4-(1-P)R_1}\right)$. Therefore, the area of ABCD will increase when D moves to the lower left $(x^* = \frac{C_4+C_5}{(1-P)R_1+R_4}\ decrease, y^* = \frac{C_2}{R_4-(1-P)R_1}\ decrease)$.

When $R_4$ increases, $s_{ABCD}$ does the same, the probability of the game system evolving to (1, 1) increases, which means that holders and users tend to choose rights-sharing approach.

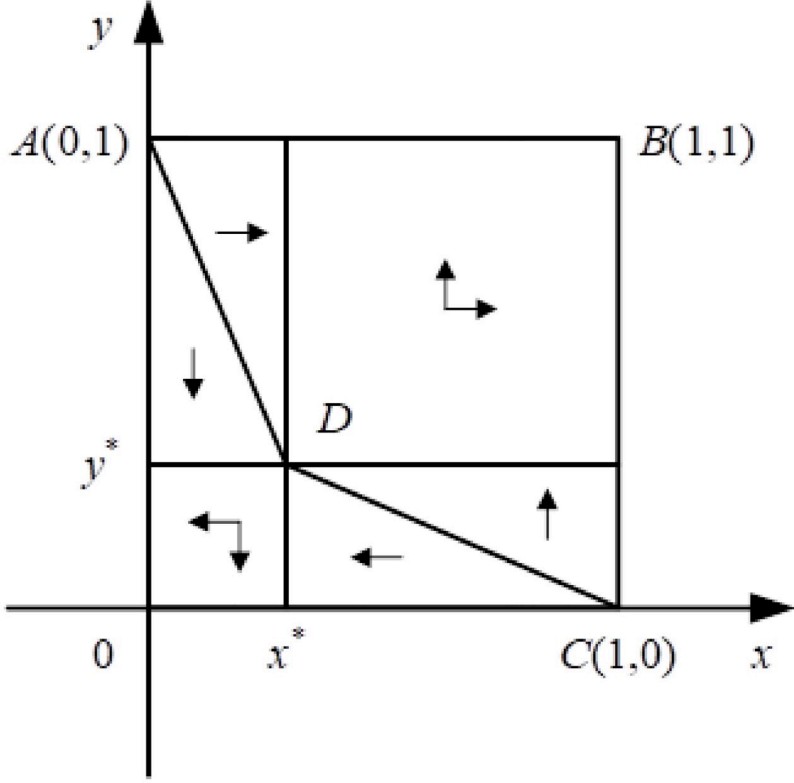

**Fig 1. Phase diagram of an evolutionary game between the two groups.** The probability of users choosing the application strategy (x) is drawn as the abscissa, and the probability of holders choosing the consent strategy (y) as the ordinate, the dynamic phase diagram of an evolutionary game between the two can be constructed.

## Validation of game simulation

The previous paper analysed the selection mechanism of users and holders in the process of transformation and utilization of traditional resources by using the theory of evolutionary game, in order to reflect the influence of parameter changes on the equilibrium state of the evolutionary game model, this paper uses Matlab R2021a programming to carry out numerical simulation, which is a clearer and more intuitive way to analyse the evolution paths of the users' and holders' strategy modes in the process of the game of transforming, utilizing and sharing of traditional resources. We verify that the above modelling analysis is realistic when "$R_4 - C_2 > (1 - p)R_1 > C_4 + C_5 - R_4$".

In the setup of our evolutionary game theory model, we have carefully chosen our initial parameters by referring to both peer-reviewed literature and real-world considerations. We believe this approach ensures our model is both theoretically sound and practically applicable. Our initial parameters are informed by a comprehensive review of similar studies in the field. This approach allows us to align our methodology with established norms and best practices. We have critically examined these studies to understand their assumptions and parameter choices, ensuring our model benefits from the collective understanding of the research community. Specific references that heavily influenced our parameter choice include Li Shasha in the "ancient books revitalisation of the status quo, problems and countermeasures preliminary exploration" for the ancient books digitization of the analysis of the lack of financial input [23]. And Liu Xingquan and Xu Yanli in "Study on the Protection Strategy of Ethnic Cultural Heritage in the Hexi Corridor Section of the Silk Road" mentioned the gap between the funds needed for the survey, collation and inheritance of intangible cultural heritage in Gansu Province and the special fund provided by the government for the protection of intangible cultural heritage [24]. In addition to academic literature, we have also taken into account the practical aspects of the situation we are modelling. We have carefully examined empirical data, field reports, and case studies, aiming to mirror as closely as possible the conditions and variables present in actual scenarios. For instance, our choice of the cost of transferring skills ($C_4$) reflects the typical range observed in relevant judicial cases [25]. Our choice of third-party funding ($R_4$) reflects the typical range observed in Guidelines for Submission of Government Funds for Safeguarding Intangible Cultural Heritage [26]. Therefore, according to the requirements of the model conditions and the considerations of the actual transaction, We assume: $C_2 = 2$, $C_4 = 5$, $C_5 = 5$, $R_1 = 40$, $R_4 = 30$, $p = 0.5$.

**The effect of differences in initial strategy choice on system evolution.** Fig 2 shows that when $R_4 - C_2 > (1 - p)R_1 > C_4 + C_5 - R_4$, users and holders have two evolutionarily stable strategies (0, 0) and (1, 1). This figure is also clear proof that long-term stable set of the evolutionary game between holders and users is (apply, consent), (not apply, not consent).

The greater the probability that the parties to the traditional resource equity game choose an affirmative strategy, the greater the probability that the traditional resource equity sharing model will be chosen. Once the two sides of the traditional resource equity game choose the application and consent strategies beyond a certain probability value, the two sides will be in a stable state in choosing the benefit-sharing mode.

**The effect of total income from the utilisation, transformation and re-creation of traditional resource rights on system evolution.** Fig 3 reflects the impact of traditional resource equity transformation and sharing of total gain on the system evolution results, the graph is the simulation results of $x$ variable with two obvious differentiation options with the change of $R_1$, and $y$ variable with the change of $R_1$ from agreeing strategy to disagreeing strategy. It can be found that different values of the total return have different impacts on the results of the system evolution. Specifically, in response to the change of the total gain, traditional resource

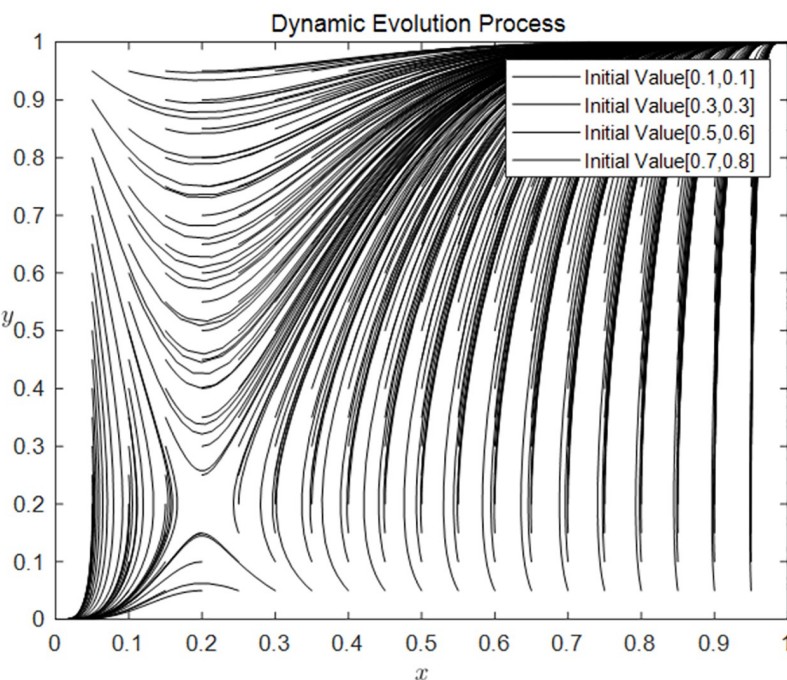

**Fig 2. Dynamic evolution process.** The probability of users choosing the application strategy (x) is drawn as the abscissa, and the probability of holders choosing the consent strategy (y) as the ordinate. The different shapes of the curves represent the strategic choices of the holders and users with different initial probabilities.

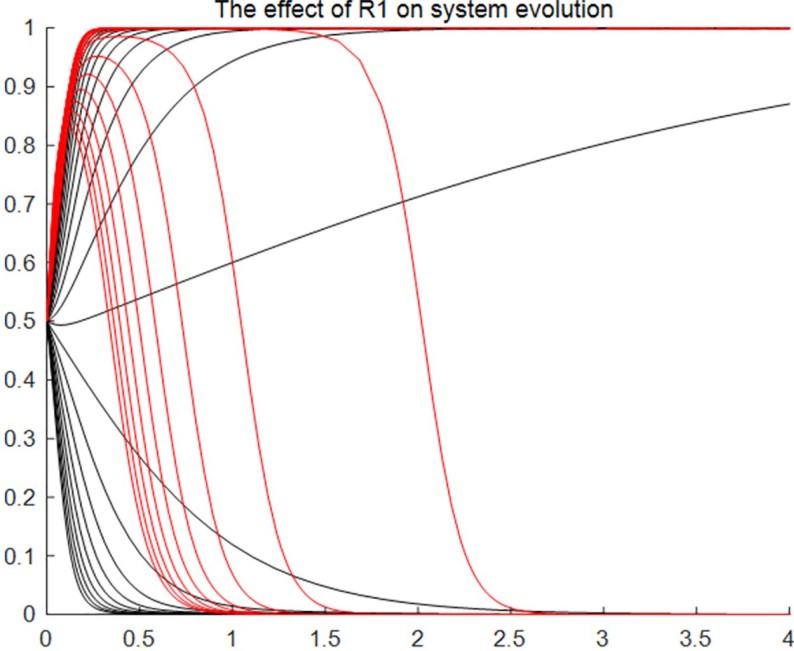

**Fig 3. The effect of total income from the utilisation, transformation and re-creation of traditional resource rights on system evolution.** Time (x) is the horizontal coordinate, and the probability of affirmative choice (y) of the two sides of the game is the vertical coordinate. The two lines of different colours represent the tendency of users and holders to choose as the total gain changes and the speed of choice.

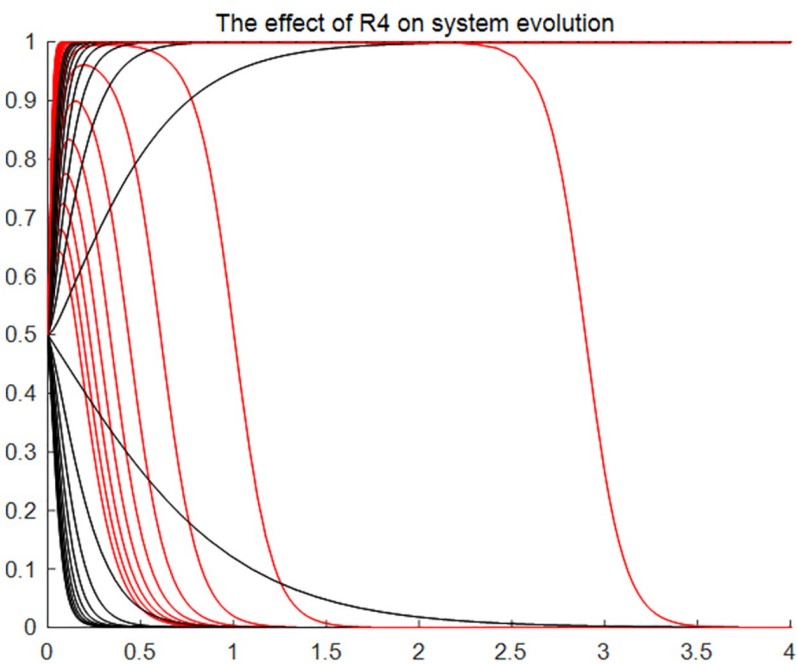

**Fig 4. The effect of the third-party funding on system evolution.** Time (x) is the horizontal coordinate, and the probability of affirmative choice (y) of the two sides of the game is the vertical coordinate. The two lines of different colours represent the tendency of users and holders to choose as well as the speed of choice in response to changes in third-party funding.

users can choose either the application strategy or the non-application strategy in the process of system evolution. However, in response to changes in total revenue, traditional resource holders will quickly choose the agree strategy at the beginning of the numerical change, but will steadily choose the disagree strategy with the growth of time and total revenue.

**The effect of the third-party funding on system evolution.** Fig 4 reflects the effect of third-party funding on the evolutionary outcomes of the system, with the graphs showing the simulation results for the $x$ variable with two distinct divergent choices as $R_4$ changes, and the $y$ variable with $R_4$ as it changes from an agreeing strategy to a disagreeing strategy. Specifically, in response to changes in third-party funding, traditional resource users can choose either an application strategy or a non-application strategy in the process of system evolution. However, in response to changes in third-party funding, traditional resource holders will quickly choose the consent strategy at the beginning of the numerical change, but will steadily choose the non-consent strategy as time and third-party funding increase.

**The effect of users' application feeholders' skill transfer cost and database building expense on system evolution.** Figs 5–7 give the results of the system evolution and the effect on the system evolution rate as application feeskill transfer cost and database building expense change, respectively. Figures show that when the initial ratio of choosing the application strategy and the consent strategy in the evolutionary game system is 0.5, the evolutionary system evolves faster towards the (application, consent) strategy with the changes of the application feeskill transfer cost and database building expense, as well as the evolution of time. The above changes reflect that the changes in application fee, skill transfer cost and database building expense have an impact on the evolution speed of traditional resource entitlement sharing and utilisation system, but do not affect the direction and outcome of the system evolution. That is,

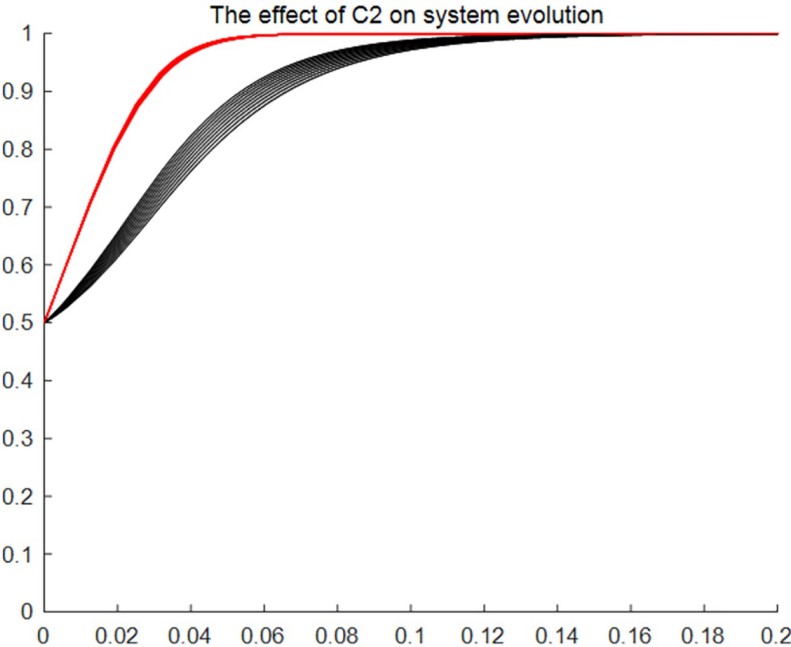

**Fig 5. The effect of users'application fee on system evolution.** Time (x) is the horizontal coordinate, and the probability of affirmative choice (y) of the two sides of the game is the vertical coordinate. The two lines of different colours represent the tendency of users and holders to choose a strategy as the application fee changes and the speed of choice.

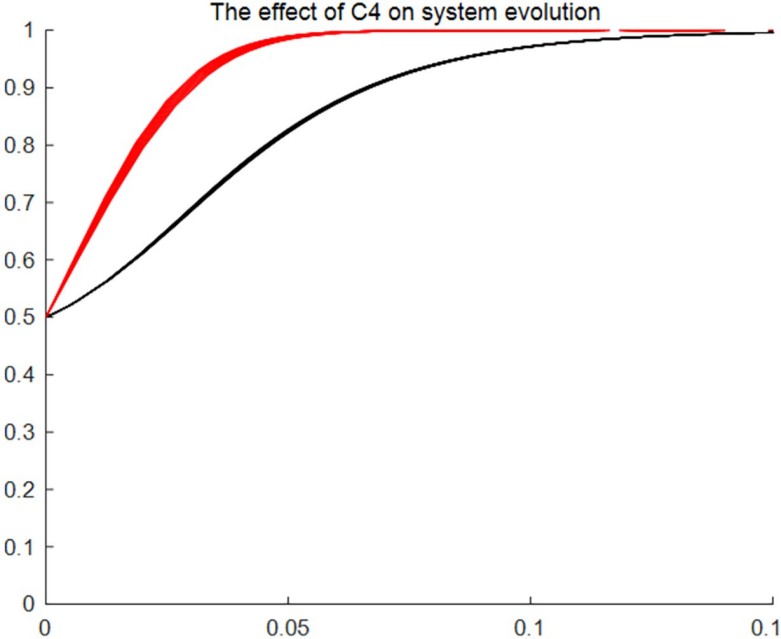

**Fig 6. The effect of holders'skill transfer cost on system evolution.** Time (x) is the horizontal coordinate, and the probability of affirmative choice (y) of the two sides of the game is the vertical coordinate. The two lines of different colours represent the tendency of users and holders to choose a strategy as the cost of skill transfer changes and the speed of choice.

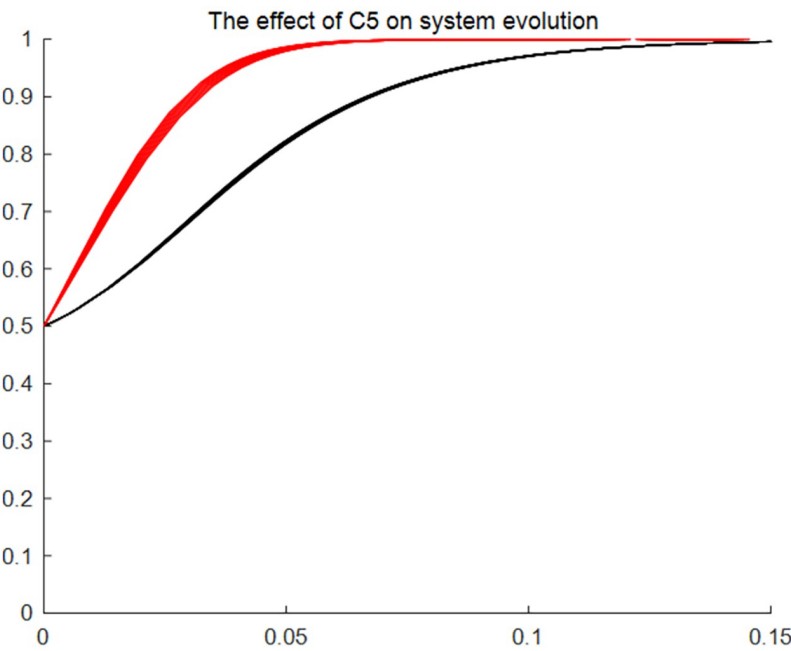

**Fig 7. The effect of database building expense on system evolution.** Time (x) is the horizontal coordinate, and the probability of affirmative choice (y) of the two sides of the game is the vertical coordinate. The two lines of different colours represent the tendency of users and holders to choose a strategy as the cost of database construction changes and the speed of choice.

regardless of the changes in application fees, skill transfer cost and database building expense, the strategy choice of traditional resource users and holders over time is the (application, consent) model.

## Results

### Evolutionary game results

The model analysis reveals the existence of a unique stable set of game processes for the conversion of traditional resource interests, i.e (not apply, not consent). At the same time, subject to certain conditions and restrictions, the long-term stable set of the evolutionary game between traditional resource holders and users is (apply, consent), (not apply, not consent).

The final choice between the parties to the conversion of traditional resource interests is closely correlated with the total income from the utilisation, transformation and re-creation of traditional resource rights($R_1$,), holders' skill transfer cost ($C_4$,), database building expense ($C_5$), users' application fee ($C_2$), and the third-party funding ($R_4$).

The choice of sharing traditional resource rights is based on users' application and holders' consent. The effectiveness condition of this path is that the difference between the third-party funding $R_4$ and users' application fee $C_2$ is greater than holders' expected benefits obtained in a benefit-sharing agreement $(1 - p)R_1$, and greater than the sum of holders' skill transfer cost $C_4$ and database building expense $C_5$ minus the third-party funding $R_4$. It is worth mentioning that this effectiveness condition is a necessary but not sufficient condition for choosing the rights-sharing approach. If the probability of users and holders choosing rights-sharing mode is to be increased, it is necessary to increase the total income from the utilisation, transformation and re-creation of traditional resource rights and the third-party funding. At the same

time, the users'application feeholders'skill transfer cost and database building expense only affects the probability of the user's choosing an application strategy and the holder's choosing a consenting strategy, and do not affect the outcome of the final choice of a sharing model by the two parties to the traditional resource interest.

## Judicial response to application and consent to sharing approach

A number of cases related to traditional resources reflect the legislative purpose of protecting rights of relevant parties. These cases also provide indirect evidence that the application and consent to sharing approach is the best choice for transforming and utilising traditional resource rights.

In Divya Pharmacy v. Union of India, the Uttarakhand High Court in India provided an innovative interpretation of the concept of benefit-sharing, ruling that an Indian company, having no foreign participation, is mandated to undertake fair and equitable benefit-sharing under the Biological Diversity Act 2002, India. The court employed a purposive interpretation, referring to the provisions of the CBD and its NP. That is, all foreign and Indian companies, institutions and individuals must obtain prior consent and approval before sharing benefits with local communities. Thus, holders' consent not only constitutes their right in benefit-sharing, but also the prerequisite for users to justify their utilisation and avoid illegal risks.

Disputes over copyright and trademark rights also provoke continuous debates. In Australia, copyright disputes over the use of indigenous artworks speak volumes about how unscrupulous businesses employ traditional resources without consent and without sharing profits. Producing indigenous artwork is a major source of income for many communities in Australia. But many non-indigenous people have engaged in producing distorted, cheap, knock-off indigenous art. In 1989, indigenous artist John Bulun Bulun (1946-2010) discovered that two of his paintings had been printed on T-shirts without his authorization. He sued for copyright infringement. Two guilty stores agreed in court to stop selling the T-shirts. The strong correlation between traditional resources and patent rights has attracted growing attention from the international society, as the neem patent case, turmeric patent and Captopril drug patent were settled. The costs of developing a new drug are prohibitive. But if researchers draw inspiration from traditional medicine, the development process can be accelerated, and the costs brought down to a great extent. But in most cases, holders of traditional medical skills do not receive due compensation. That is why disputes over neem tree, basmati rice, and turmeric have sparked strong protests from Indian companies and public interest groups against the unscrupulous exploitation of traditional resources.

In China, due to inadequate legal frameworks for protecting traditional resource rights, courts at all levels rely on existing laws and regulations to protect traditional skills and expressions. For example, Bai Xiu'e v. State Post Bureau and Postage Stamp Printing Bureau of China Post Group for copyright infringement. After analysing differences between traditional knowledge and folk art, and between group creation and individual inheritance, Beijing Higher People's Court determined that paper-cut artist Bai Xiu'e was qualified as a subject of litigation, which means that Bai Xiu'e can claim her individual right to "re-create existing paper-cut forms", but not collective right to traditional elements such as "paper-cutting techniques" and "traditional paper-cutting" [27]. Another case, Chen Qihua and other 116 villagers from Xiamiao No. 1 Village, Qiaotou Town, Datong Hui and Tujia Autonomous County, Qinghai Province v. Chai Yukui, Chai Mingxiao, Gansu Provincial Audio-visual Publishing House, and Qinghai Qiaojia Audio-visual Co., Ltd. for infringement on Community Fire, a traditional spring festival. Although the right jointly claimed by the 117 villagers was limited to performance right, a private interest, which cannot be extended to traditional elements of folk

literature and art, namely, the form of Community Fire performances and singing tunes, joint litigation protected individual interests, and preserved the folk-art tradition to some extent.

These cases show that viable ways of transforming traditional resources include users' respect for holders' benefit-sharing right, improvement in traditional treatment, and effective solution to resolving indigenous art and music misappropriation. The right strategy can generate three benefits: indigenous holders' ready consent to others' uses and transformation of traditional resources, transfer of authentic skills, and building of databases to present local traditional resources; reduction in costs incurred by illegal commercialization, and by brand building of interest groups; and wide recognition and continuous funding of third-party organizations for the inheritance project. We believe that when transforming traditional resources, stakeholders can adopt an open sharing mode that facilitates users' application and holders' consent to maximize overall benefits.

## Discussion

Based on four major breakthroughs represented by WIPO-GRTKF, and the optimal path indicated by the evolutionary game model, we provide a solution that fits the Chinese context, combining justified utilisation and statutory licensing to protect multiple rights arising from the transformation and utilisation of traditional resources. We also propose that holders and users' rights, obligations and interests be specified and safeguarded, and China Traditional Resources Management Association and traditional resource databases be created and maintained.

### Specifying multiple rights under a hybrid legislative system to protect private and public interests

When transforming traditional resources, decision-makers are expected to consider multiple rights covering both public and private dimensions. Accordingly, the sharing of transformed traditional resource rights should include the safeguard of traditional resource rights, and the prevention of unwarranted use. Holders have positive powers to utilise, share and dispose the resources, as well as defensive powers to prevent users from unwarranted possession and utilisation. Specifically, the former is a private right which is protected by the legal system of IPRs, while the latter is a public right, protected by competition law and administrative law. In other words, a hybrid theory is suitable for the protection of traditional resource rights, a source of scholarly contention mentioned above. We believe that a hybrid legal system incorporating both private law and public law with IPRs at the core is indispensable for the protection of traditional resource rights. This is especially true for the sharing process.

We find that the content produced by users through the transformation of traditional resources is very similar to the content related to the neighbouring rights, and both rights are non-original. Therefore, when designing the sharing system of traditional resource rights, decision-makers can draw on the neighbouring rights system under the Copyright Law. In the Copyright Law of the People's Republic of China, neighbouring rights include those enjoyed by the publisher, performer, recorder, and radio and television organizer. The subject of traditional resource rights, or holders of traditional resources, has original rights similar to copyright, while users of the resources enjoy derivative property rights similar to neighbouring rights. Thus, it is feasible to adopt the neighbouring rights system to proscribe rights and obligations of traditional resource holders and users.

If the neighbouring rights system is adopted, users of traditional resources enjoy the rights to use traditional resources, re-create them, build brands, and organize fund-raising events. Our empirical model is further supported by the following facts: Users obtain a source of

income from using and re-creating traditional resources ($R_1$), from building brands based on authentic resources by selecting application ($R_2$), and from the third-party funding under the rights-sharing agreement ($R_4$). At the same time, users need to fulfil obligations of origin disclosure, informed consent, benefit-sharing, the preservation of spiritual core of traditional resources, respect for holders' beliefs, and no unwarranted possession. This confirms our model that users need to pay obligation-induced cost ($C_1$), application fee ($C_2$) and illegal cost ($C_3$). Regarding holders, they can obtain a source of income from transforming traditional resources, from developing local skills, and from organizing fund-raising events. This is also supported by our model. That is, holders can obtain a certain percentage of income $R_1$ from the utilisation, transformation and re-creation of traditional resources, $R_3$ from the utilisation of local skills, which contributes to their inheritance and development, and the third-party funding $R_4$ under a rights-sharing agreement. But holders also need to bear the cost of transferring skills ($C_4$)and building databases ($C_5$).

To sum up, we advocate for a hybrid system that combines the incentive of IPRs as private rights, and the safeguard of public rights. In the dual subject system, both users and holders enjoy multiple rights to share, be respected, benefit, consent, use, transform and re-create traditional resources.

## Establishing a system combining justified utilisation and statutory licensing based on open sharing principle

Given that China has no robust legal system dedicated to traditional resources, we propose that special laws and stand-alone regulations be formulated to strengthen the system of transforming and utilising traditional Chinese resources. First and foremost, based on international practices, an autonomous and open sharing approach should allow holders of original rights to apply different principles to domestic users and foreign ones. For the former, justified utilisation is preferred. For the latter, statutory licensing is optimal.

To be more specific, domestic users need to obtain informed consent from holders and other stakeholders. Under the framework of IPRs, individual and corporate users are allowed to access and utilise traditional resources for appreciation, classroom demonstration, and scientific research [28]. Complying with statutory rights principle [29], applicants should provide information on patent source. Justified utilisation principle is applied to domestic users. When they use national traditional resources, neighbouring communities do not need to pay utilisation fees. But domestic users should disclose origin and obtain informed consent.

For foreign users, we propose to conclude a licensing agreement based on the statutory licensing system for benefit sharing. While encouraged to transform traditional Chinese resources, they should comply with the statutory licensing principle, and fulfil benefit-sharing obligation, apart from informed consent and origin disclosure. The benefit sharing system enables users and holders to share the benefits brought about by the development and utilisation of traditional resources. To restrict the monopoly of IPRs, foreign users who access and utilise traditional resources according to the statutory licensing term should, in line with their IPRs, grant Chinese traditional resource holders non-exclusive and free licenses. In practice, this principle has been applied to protect Chinese folk literature and art. This achievement convinces some scholars that systems such as benefit-sharing, statutory licensing and use registration are pragmatic solutions to clarifying stakeholders' rights and obligations [30]. Therefore, both statutory licensing and benefit-sharing are viable approaches to transforming traditional resources and sharing rights.

Costa Rica sets a good example of using the Merck-INBio Agreement to share the benefits associated with traditional resources. The National Biodiversity Institute (INBio), a non-profit

scientific organization created by the government of Costa Rica, signed an agreement with Merck, a multinational science and technology company. The two sides agreed that Merck pays a one-time fee to INBio [31] which provides samples of extracts from resources in Costa Rica, and that Merck has the exclusive rights to study these samples [32], and retain the patents to any drugs developed using the samples. Merck should pay INBio proceeds in proportion to sales. This is a typical benefit-sharing approach under the statutory licensing system. The approach adopted by the International Cooperative Biodiversity Groups (ICBG) is also praise-worthy. A specialised agency receives funds from the U.S. government for collecting samples from source countries and sends them back for research and development. To share interests, the source countries can negotiate with the agency on an as-needed basis. The benefit-sharing duration varies from long-term to short-term. The form of benefits usually includes about 1%-3% of license fee, scientist training programme, and equipment donation [33]. Therefore, the benefit-sharing mode for foreign users under the statutory licensing system can be a "one-off payment + proceeds-sharing" or "on an as-needed basis", and the form of benefit can be pro-ceeds, skill transfer, scientist training programme and equipment donation. In other words, when utilising, transforming, and re-creating traditional resource rights, stakeholders can obtain various forms of benefits, without compromising the purpose of open sharing and mutual advantage.

## Integrating holders and users' rights, obligations, and interests

First, it is necessary to establish a system that safeguards holders' consent to the transforma-tion, and clarifies rights, obligations, and interests. The system should cover holders' prior informed consent, agreement to benefit-sharing terms, and rights to agree with, benefit from and participate in the transformation and receive funds. In the process of transforming, utilis-ing, and re-creating traditional resource rights, holders can share the total income $R_1$, obtain additional benefits from users $R_3$, and receive the third-party funding $R_4$. At the same time, users should bear illegal cost $C_3$ if they do not obtain holders' consent. It is critical to ensure the access to traditional resources and fair share of benefits under agreed terms, safeguard holders' fundamental rights, and avoid unauthorised use in the process of transforming and sharing traditional resource rights. Holders and users should refrain from choosing the following three paths: (not apply, consent), (apply, not consent) and (not apply, not consent).

Second, users should be incentivized to transform traditional resources in accordance with laws and regulations, and granted with the rights to apply for use, re-create, share benefits, build brands, and sponsor events. In other words, users should be enabled to share benefits from the use of traditional resources with holders and incentivized to seek re-creation path to develop traditional resources. That is because the rights-sharing approach allows users to share three forms of benefit, namely, total income from transforming and re-creating traditional resource rights ($R_1$), branding income from using authentic resources ($R_2$), and third-party funding ($R_4$), and bear two types of cost, i.e. input cost for transforming and re-creating ($C_1$), and application fee ($C_2$). Users include technology development companies and social organi-zations. Through legal guarantees and policy incentives of justified utilisation and statuary licensing, holders can empower or cooperate with users to transform and utilise authentic tra-ditional resources and increase the efficacy of the whole process.

## Building a traditional resource management association and databases

**Forming associations.** A specialized association is a prerequisite for the inheritance of authentic traditional resources. Support measures include a detailed introduction about how to access traditional resources, and what requirements the holders should meet, and what

procedures the association will follow. To make informed decisions, members of the association come from national-level administrative authorities, private funding groups, research institutions, holders of traditional resources, and stakeholders' representatives. Their job responsibilities include formulating and assisting in the implementation of protection and development policies for traditional resources, following procedures of justified utilisation and authorised licensing, providing database registration standards, offering contractual templates for transformation and rights-sharing, establishing a contract filing mechanism, and granting rights to users on behalf of holders at all levels. Community-level chapters can be established to offer accessible services. The ultimate aim of the association is to reduce users' application fee $C_2$, skill transfer cost $C_4$, and database building expense $C_5$, while increasing the third-party funding $R_4$.

**Building databases.** The association can play a coordinating role in building nationwide databases, and in introducing measures to update and exchange data and safeguard security. All this is to facilitate users' application and holders' consent, form close partnerships, balance between interests and sustainable development, and access quality services.

To build databases in China, decision-makers can draw inspiration from international practices, and theories well received by global audiences. WIPO/GRTKF/IC/47/17 has a central theme, "Joint recommendation on the use of databases for the defensive protection of genetic resources and traditional knowledge associated with genetic resources" [34]. The document states, "...searchable databases under the proposed system should be in the possession of, and maintained by, each participating WIPO member state. The database will be composed of a WIPO portal site as well as databases of WIPO member states, which are linked to this portal site." Developing countries should leverage digital technologies to build traditional resource databases and provide financial support. Decision-makers should be fully aware that databases lower the cost of transforming and sharing traditional resources, firing holders' enthusiasm to choose the rights-sharing approach. Reducing holders' cost to build databases also gives more incentives for users to choose the same approach. To translate this vision into reality, decision-makers should pay special attention to information declaration and database maintenance.

First, we suggest a declaration system be created for holders of traditional resource rights. Things to be declared should include TK, F and GR. Decision-makers can refer to UNESCO's First Proclamation of Masterpieces of the Oral and Intangible Heritage of Humanity, and adapt it to local conditions, without disclosing trade secrets. Given that multiple stakeholders are involved in transforming, utilising and sharing traditional resources, the data should be reviewed by traditional resource management association authorised by government authorities which commission an expert panel or a specialised appraisal body to validate the data before the entry into databases. At the same time, a plan about how to protect and transform traditional resources, corresponding budget, authorization brief about justified utilisation, and rights-sharing terms should be provided for the association to exercise power and distribute interests. More emphasis should be given to fund-providing third parties whose identity should be diversified. They should be incentivized to increase the source of investment and seek more investors who can increase the third-party funding $R_4$ for the transformation and utilisation of traditional resources, thus increasing the probability of users and holders choosing the rights-sharing approach.

Second, we propose that operation and maintenance of databases be prudent. India's Traditional Knowledge Digital Library (TKDL) and Ecuador's Secret Data Bank set an outstanding example. The TKDL integrates all essential elements of traditional Indian Unani, siddka, yoga and naturopathy. Patent examiners have easy access to source information related to Indian Traditional Medicine [35]. It is this organization that played a crucial role in revoking the

neem tree patent case in 1994. The success lies in experts' thorough review of patents to avoid erroneous grant, and comprehensive recording of Indian medicines such as Unani, siddka, yoga and naturopathy.

In the process of collecting traditional resources and reviewing their patent information, free-rider problem and careless disclosure are major challenges that need to be addressed. Ecuador stores its TK in the Secret Data Bank, and entrusts EcoCiencia, a non-governmental organization, to file and register traditional resource information [36]. Ecuador's practice inspires us that when building databases, e-libraries and e-maps, decision-makers should conduct a preliminary examination of patent, trademark and copyright application from natives and foreigners, avoid erroneous grant of patents, and negotiate with any third party who wants to obtain trade secret. Special attention should also be paid to unintended disclosure of data or materials because they are not perceived as the object of IPRs. Therefore, we suggest that, in the process of building China's traditional resource databases, only essential information be disclosed, nationwide supervision conducted, and application and consent cross-checked. When domestic users of transformed traditional resources apply to a neighbouring community, and are qualified for justified utilisation, they can use the resources freely based on informed consent. For domestic cross-community users, if they meet certain conditions, they can use the resources based on informed consent, and be given tax deduction for the income obtained from the transformation. But they should share a certain percentage of income with holders. For foreign users of transformed traditional resources, if they meet certain conditions, and pass security review, they can use the resources after obtaining informed consent, and give a certain percentage of income to holders.

## Conclusion

Basically, through the evolutionary game model to choose the rights-sharing approach, it has declared that the final choice between the parties to the conversion of traditional resource interests is closely correlated with the total income from the utilisation, transformation and re-creation of traditional resource rights, holders' skill transfer cost, database building expense, users' application fee, and the third-party funding. Concurrently, as the negotiation of new WIPO-GRTKF is unfolding, international game system for sharing rights arising from the transformation of traditional resources is being reshaped. When transforming traditional resources, to increase the probability of users and holders choosing a rights-sharing model the parties need to increase the total benefits from the conversion and utilisation of traditional resources, seek third-party financial support. We believe that the Chinese approach will inspire international negotiators to diversify rights and increase the accessibility of the sharing process.

## Author Contributions

**Conceptualization:** Dong Zhang.

**Data curation:** Rui Huang, Chaoran Lin.

**Formal analysis:** Dong Zhang.

**Investigation:** Rui Huang, Chaoran Lin.

**Methodology:** Rui Huang, Chaoran Lin.

**Project administration:** Dong Zhang.

**Supervision:** Dong Zhang.

**Validation:** Rui Huang, Chaoran Lin.

**Writing – original draft:** Dong Zhang.

**Writing – review & editing:** Dong Zhang, Rui Huang, Chaoran Lin.

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
