## [Decision Letter · Decision Letter 0]

15 Sep 2023

PONE-D-23-24767

WIPO Initiative of Transforming Traditional Resources and Sharing Rights: An Evolutionary Game Analysis and A Chinese Context

PLOS ONE

Dear Dr. Zhang,

Thank you for submitting your manuscript to PLOS ONE. After careful consideration, we feel that it has merit but does not fully meet PLOS ONE’s publication criteria as it currently stands. Therefore, we invite you to submit a revised version of the manuscript that addresses the points raised during the review process.

We look forward to receiving your revised manuscript.

Kind regards,

Don Sasikumar

Academic Editor

PLOS ONE

Journal Requirements:

Additional Editor Comments:

Dear Authors

We have completed the review process of your article. Please read the reviewers' comments and revise your article in light of their comments.

Reviewers' comments:

Reviewer's Responses to Questions

**Comments to the Author**

1. Is the manuscript technically sound, and do the data support the conclusions?

Reviewer #1: Partly

Reviewer #2: Partly

2. Has the statistical analysis been performed appropriately and rigorously? 

Reviewer #1: I Don't Know

Reviewer #2: No

3. Have the authors made all data underlying the findings in their manuscript fully available?

Reviewer #1: No

Reviewer #2: No

4. Is the manuscript presented in an intelligible fashion and written in standard English?

Reviewer #1: Yes

Reviewer #2: No

5. Review Comments to the Author

Reviewer #1: As the WIPO-GRTKF negotiations unfold, the international framework for sharing rights, stemming from the transformation of traditional resources, is undergoing revision. This study introduces an empirical model detailing how resource holders and users might collaborate to adapt traditional resources and share associated rights. The model suggests that international negotiators can diversify rights and base their decisions on empirical evidence regarding the rights-sharing process.

In this manuscript, the authors categorize the rights related to the transformation and utilization of traditional resources under four main perspectives:

A Robust Structure: Valuing Respect and Sharing

Stakeholder Game Relations: Holders vs. Users

Components of Subject: Differentiating Holders from Users

Game Relationships: A Detailed Examination

Moreover, encapsulating these perspectives, the "Evolutionary Game Reasoning and Analysis" methodology is employed. The authors validate scenarios for managing the rights originating from resource transformations using specified equations (Eq.1~Eq.5).

The game simulation indicates that the access approach to rights sharing, while necessary for traditional resource transformation, doesn't suffice. This is because both users and holders possess the autonomy to make strategic decisions (as depicted in Figure 2, Page 10).

To put it briefly, this manuscript offers insights into rights sharing and management derived from the transformation of traditional resources, all grounded in an empirical model. Nonetheless, a more objective and quantitative assessment of the proposed model appears warranted. The authors might consider furnishing the empirical model's experimental results to bolster the assertions. This could involve broadening the scope beyond the instances in Table 2 and integrating a more diverse set of factors.

Furthermore, the formatting on Page 7 requires attention, particularly the consistent indentation of the first bracketed number in each section.

Reviewer #2: abstract is too wordy.make it simple to the readers

no references for the statements in the introduction

can include more related latest references

references are not well arranged. It shall be arranged as per the format given

Ensure all the reference are cited.

when it is related with chinese context, then why the authors want to take the cases in india

Results and Discussion missing, some comparison with the proposed approach need to be added

The flow of the paper should be improved.

The language in the paper should be improved.

Conclusion should clearly state the outcome.

6. PLOS authors have the option to publish the peer review history of their article (what does this mean?). If published, this will include your full peer review and any attached files.

Reviewer #1: No

Reviewer #2: No

---

## [Author Response · Author response to Decision Letter 0]

8 Dec 2023

The authors appreciate the opportunity to submit a revised draft of the manuscript “WIPO Initiative of Transforming Traditional Resources and Sharing Rights: An Evolutionary Game Analysis and A Chinese Context” for publication in PLOS ONE. We appreciate the time and effort that the academic editor and the two reviewers dedicated to providing comments and suggestions, which have helped us implement valuable improvements to our paper. We have addressed all the journal requirements and incorporated most of the suggestions made by the reviewers in the revised paper, detailed below in point-by-point responses. We have attached the revised paper with changes highlighted in addition to a “clean” copy. We will be willing to make further revisions if necessary. 

Journal Requirements:

Author response:

We have formatted the revised paper following PLOS ONE style requirements, using the PLOS ONE style templates. 

Author response:

Below is our revised Data Availability statement.

We have made the Matlab running code for the simulation of the evolutionary game results available in our GitHub project. This will allow others to access the same data and simulation code analysed in this study. These data can be viewed on the GitHub database website via the link below:

https://github.com/huangruiipr/Traditional-Resource-Evolution-Game.git

Data can also be requested from the corresponding author upon reasonable request.

Author response:

Regarding the request for a full ethics statement in the 'Methods' section of our manuscript, I would like to clarify that our study did not involve any activities that would require ethical review and approval.

Our research did not involve human participants or animals, and no personal or sensitive data was used. Therefore, it was not necessary to seek approval from an Institutional Review Board (IRB) or ethics committee.

However, we understand the importance of ethics in research and assure you that our study was conducted in accordance with the general ethical guidelines of our field.

We have updated the 'Methods' section of our manuscript to clearly state this information and to avoid any potential confusion for readers.

Thank you again for bringing this to our attention. We appreciate your efforts in helping us improve our manuscript.

Reviewers' comments:

Reviewer's Responses to Questions

Comments to the Author

1. Is the manuscript technically sound, and do the data support the conclusions?

Reviewer #1: Partly

Reviewer #2: Partly

2. Has the statistical analysis been performed appropriately and rigorously?

Reviewer #1: I Don't Know

Reviewer #2: No

3. Have the authors made all data underlying the findings in their manuscript fully available?

Reviewer #1: No

Reviewer #2: No

4. Is the manuscript presented in an intelligible fashion and written in standard English?

Reviewer #1: Yes

Reviewer #2: No

5. Review Comments to the Author

Reviewer #1: As the WIPO-GRTKF negotiations unfold, the international framework for sharing rights, stemming from the transformation of traditional resources, is undergoing revision. This study introduces an empirical model detailing how resource holders and users might collaborate to adapt traditional resources and share associated rights. The model suggests that international negotiators can diversify rights and base their decisions on empirical evidence regarding the rights-sharing process.

In this manuscript, the authors categorize the rights related to the transformation and utilization of traditional resources under four main perspectives:

A Robust Structure: Valuing Respect and Sharing

Stakeholder Game Relations: Holders vs. Users

Components of Subject: Differentiating Holders from Users

Game Relationships: A Detailed Examination

Moreover, encapsulating these perspectives, the "Evolutionary Game Reasoning and Analysis" methodology is employed. The authors validate scenarios for managing the rights originating from resource transformations using specified equations (Eq.1~Eq.5).

The game simulation indicates that the access approach to rights sharing, while necessary for traditional resource transformation, doesn't suffice. This is because both users and holders possess the autonomy to make strategic decisions (as depicted in Figure 2, Page 10).

Author response:

We greatly appreciate the reviewer’s recognition of the strengths and potentials of our study.

To put it briefly, this manuscript offers insights into rights sharing and management derived from the transformation of traditional resources, all grounded in an empirical model. Nonetheless, a more objective and quantitative assessment of the proposed model appears warranted. The authors might consider furnishing the empirical model's experimental results to bolster the assertions. This could involve broadening the scope beyond the instances in Table 2 and integrating a more diverse set of factors.

Author response:

In the second part of the article, We have carried out a more objective quantitative assessment of the model. In the third section of the second part of the article, We make systematic improvements and additions to the simulation and validation of the model. We highlight the rationale for not requiring a database for the evolutionary game model on page 9 of the article. In the simulation section we have considered and analysed all the factors affecting the conversion and sharing of traditional resource rights and interests, and we have explained the criteria and sources for choosing the values (see page 9, para 1). At the same time, the graphs produced in this study were optimised. We redrew the diagram of the evolutionary game process of traditional resource rights conversion and sharing(see Fig 2), and added the analysis of the influence of different factors on the results of the evolutionary game (see page 9,10,11) .

Among them，Figure 3 reflects the impact of traditional resource equity transformation and sharing of total gain on the system evolution results, the graph is the simulation results of x variable with two obvious differentiation options with the change of , and y variable with the change of from agreeing strategy to disagreeing strategy (see page 9). Figure 4 reflects the effect of third-party funding on the evolutionary outcomes of the system, with the graphs showing the simulation results for the x variable with two distinct divergent choices as changes, and the y variable with as it changes from an agreeing strategy to a disagreeing strategy (see page 10). Figures 5, 6 and 7 give the results of the system evolution and the effect on the system evolution rate as application fee，skill transfer cost and database building expense change, respectively (see page 10).

Furthermore, the formatting on Page 7 requires attention, particularly the consistent indentation of the first bracketed number in each section.

Author response:

Thank you to the reviewers for their thoughtful suggestions. We have corrected the formatting where it was incorrect and this has resulted in consistent indentation of the numbers in brackets in each section.

Reviewer #2: abstract is too wordy. make it simple to the readers

Author response:

Abstracts of articles have been streamlined. The revisions have been highlighted in the revised paper. In particular, the background to the writing of the article and the use of evolutionary gaming methods are presented more succinctly.

no references for the statements in the introduction

Author response:

We have added references in the introduction. Reference I is a new addition.

can include more related latest references

references are not well arranged. It shall be arranged as per the format given

Ensure all the reference are cited.

Author response:

We have cited the latest references and organised the references in the format given. Revised the labelling in the article where references were not added. In particular, the formatting of references 2, 15, 22 and 29 has been corrected.

when it is related with chinese context, then why the authors want to take the cases in india

Author response:

In the third part of the article, examples are also given to illustrate relevant cases of conservation of traditional resources in Australia. China context exploration need a comparative study with other countries which also being rich in traditional resources. For this purpose, authors quote some representative cases not only Divya Pharmacy v. Union from India but also the case Aboriginal artworks from Australia.

Results and Discussion missing, some comparison with the proposed approach need to be added

Author response:

We have added Results and Discussion. It focuses on the comparison of the stability of evolutionary game models under different selection strategies, with an emphasis on the key factors affecting the stability of the model, and how the probability of strategy selection by holders and users affects the outcome of the transformation of traditional resource interests.

The flow of the paper should be improved.

The language in the paper should be improved.

Author response:

Many thanks to the reviewers for their suggestions on the logic and language of the article. We have improved the flow of the article and endeavoured to improve the English presentation. The sections and structure of the article have been changed in accordance with the requirements of the journal and are structured as Introduction, Materials and Methods, Results, Discussion and Conclusion. The first chapter was shortened to give more emphasis to the innovation of an evolutionary game analysis of the conversion process of traditional resource rights and interests.

Conclusion should clearly state the outcome.

Author response:

We modified the presentation of the paper's conclusion to make it a more direct statement of the results. In the conclusion, the role of the evolutionary game model for the transformation and sharing of traditional resource rights is directly stated against the "Results and Discussion" in Part III of the paper. When transforming traditional resources, to increase the probability of users and holders choosing a rights-sharing model，the parties need to increase the total benefits from the conversion and utilisation of traditional resources , seek third-party financial support(see page 17).

6. PLOS authors have the option to publish the peer review history of their article (what does this mean?). If published, this will include your full peer review and any attached files.

Do you want your identity to be public for this peer review? For information about this choice, including consent withdrawal, please see our Privacy Policy.

Reviewer #1: No

Reviewer #2: No

---

## [Decision Letter · Decision Letter 1]

15 Jan 2024

WIPO Initiative of Transforming Traditional Resources and Sharing Rights: An Evolutionary Game Analysis and A Chinese Context

PONE-D-23-24767R1

Dear Dr. Zhang,

We’re pleased to inform you that your manuscript has been judged scientifically suitable for publication and will be formally accepted for publication once it meets all outstanding technical requirements.

Kind regards,

Don Sasikumar

Academic Editor

PLOS ONE

---

## [Editor Report · Acceptance letter]

25 Mar 2024

PONE-D-23-24767R1 

PLOS ONE

Dear Dr. Zhang, 

I'm pleased to inform you that your manuscript has been deemed suitable for publication in PLOS ONE. Congratulations! Your manuscript is now being handed over to our production team.

Kind regards, 

on behalf of

Dr. Don S 

Academic Editor

PLOS ONE